# Fragile X Syndrome Caused by Maternal Somatic Mosaicism of *FMR1* Gene: Case Report and Literature Review

**DOI:** 10.3390/genes13091609

**Published:** 2022-09-08

**Authors:** Maria Jose Gómez-Rodríguez, Montserrat Morales-Conejo, Ana Arteche-López, Maria Teresa Sánchez-Calvín, Juan Francisco Quesada-Espinosa, Irene Gómez-Manjón, Carmen Palma-Milla, Jose Miguel Lezana-Rosales, Ruben Pérez de la Fuente, Maria-Luisa Martin-Ramos, Manuela Fernández-Guijarro, Marta Moreno-García, Maria Isabel Alvarez-Mora

**Affiliations:** 1Genetic Service, Hospital Universitario 12 de Octubre, 28041 Madrid, Spain; 2UdisGen-Unidad de Dismorfología y Genética, Hospital Universitario 12 de Octubre, 28041 Madrid, Spain; 3Network Center for Biomedical Research in Cancer (CIBERONC), 28029 Madrid, Spain; 4Internal Medicine Department, Hospital Universitario 12 de Octubre, 28041 Madrid, Spain; 5Department of Biochemistry and Molecular Genetic, Hospital Clínic de Barcelona, 08036 Barcelona, Spain

**Keywords:** *FMR1*, *FMR1* gene deletion, fragile X syndrome, mosaicism, rare FXS mutation

## Abstract

Fragile X syndrome (FXS) is caused by an abnormal expansion of the number of trinucleotide CGG repeats located in the 5′ UTR in the first exon of the *FMR1* gene. Size and methylation mosaicisms are commonly observed in FXS patients. Both types of mosaicisms might be associated with less severe phenotypes depending on the number of cells expressing FMRP. Although this dynamic mutation is the main underlying cause of FXS, other mechanisms, including point mutations or deletions, can lead to FXS. Several reports have demonstrated that *de novo* deletions including the entire or a portion of the *FMR1* gene end up with the absence of FMRP and, thus, can lead to the typical clinical features of FXS. However, very little is known about the clinical manifestations associated with *FMR1* gene deletions in mosaicism. Here, we report an FXS case caused by an entire hemizygous deletion of the *FMR1* gene caused by maternal mosaicism. This manuscript reports this case and a literature review of the clinical manifestations presented by carriers of *FMR1* gene deletions in mosaicism.

## 1. Introduction

Fragile X syndrome (FXS, OMIM #300624) is the leading monogenic form of intellectual disability (ID) and autism spectrum disorder (ASD) affecting around 1:5000 males and 1:8000 females [1]. Developmental delay is commonly noticed during the first years of life, particularly with language delays. FXS patients can present with severe behavioral alterations, including hyperactivity, impulsivity, and anxiety, in addition to poor language development and seizures. Males are generally more affected, while females tend to present with a less severe phenotype by the activation index of the second unaffected X chromosome [2].

FXS is caused by the loss of function of the fragile X messenger ribonucleoprotein 1 (*FMR1* *300805) gene located at Xq27.3, which encodes the fragile X messenger ribonucleoprotein (FMRP) [3]. FMRP is an RNA-binding protein that regulates the translation of approximately 4% of fetal brain mRNA. This protein is expressed in neural stem cells and probably in all neuronal and glial cell types. FMRP forms large ribonucleoprotein complexes which regulate the transport, translation, and metabolism of mRNAs, a crucial activity for appropriate neuronal development, synaptic connectivity and plasticity, and dendritic architecture [4].

Nearly all FXS cases are caused by a CGG repeat expansion in the 5′ UTR of the *FMR1* gene. FXS patients carry more than 200 CGG repeats, referred to as “full-mutation” alleles (FM), which generally lead to hypermethylation of the repeat itself and the promoter region and transcriptional silencing of the gene [5]. This dynamic mutation is reported in approximately 95–99% of FXS cases. However, in some cases, rare mutations leading to FXS have been reported, including point mutations (variants in promoter regions, splice sites or missense, nonsense, and frameshift variants) as well as copy number variants (CNVs) [6]. Currently, there are 69 different variants reported in the HGMD database (HGMD^®^ Professional 2022.2) as disease-causing mutations not related to CGG repeat expansion. Among these, deletions represent more than 60% (43/69), including five partial deletions, 25 entire gene deletions, and 13 deletions encompassing the 5′ UTR CGG repeat region without affecting coding sequences. Several reports have demonstrated that de novo deletions including the entire or a portion of the *FMR1* gene end up with the absence of FMRP and, thus, can lead to the typical clinical features of FXS associated with FM alleles.

Here, we report an FXS patient with an entire deletion of the *FMR1* gene inherited from his asymptomatic mother, who was a mosaic carrier of the deletion. Additionally, we review the current literature and summarize the clinical manifestations of FXS carriers of an *FMR1* gene deletion in mosaicism.

## 2. Materials and Methods

### 2.1. Subject

The patient (II.1) is a 39-year-old Spanish man presenting with nonprogressive ID with a peculiar phenotype, but not specific to FXS (Figure 1). Family history was negative for ID. He was referred for genetic testing to the Genetic Service of Hospital Universitario 12 de Octubre.

The family provided written informed consent for the testing and use of their clinical and genetic data. The study was carried out in accordance with the Declaration of Helsinki (2013).

### 2.2. Copy Number Variant Analysis

Whole peripheral blood from the patient and the relatives was collected in a heparin tube and sent to the Genetic Service of Hospital 12 de Octubre for chromosomal microarray (CMA) analysis. DNA from all the samples was extracted using conventional procedures. A 60K CGH array (60 K KaryoNIM^®^, NIMGenetics, Madrid, Spain) was performed following the manufacturer’s instructions. This platform provides a resolution of 350 kb in the backbone and of 100 kb in targeted regions. The hg19 and ADM-2 algorithms (Aberration Detection Method 2) were used. Analysis and interpretation were performed using Cytogenomics (v.4.0.3.12, Agilent, Santa Clara, CA, USA). A threshold of ≥ 5 consecutive probes was established to consider a CNV. The CNVs found in the patient were analyzed in comparison with public databases (DGV, DECIPHER) and classified as outlined elsewhere [7].

The genetic alteration was confirmed by means of a multiplex ligation-dependent probe amplification (MLPA) assay using SALSA MS-MLPA Probemix ME029-FMR1/AFF2-B3 at an external laboratory. This MLPA SALSA provided information on copy number variants and methylation status of the *FMR1* and *AFF2* genes.

### 2.3. Molecular Analysis of the FMR1 CGG Repeat Locus

To characterize the mosaicism, the patient’s mother (II.4) was also analyzed by means of triplet repeat primed PCR (RTP PCR) using an AmplideX^®^ FMR1 PCR commercial kit (Asuragen, Austin, TX, USA). The sequences were analyzed with Gene Mapper™.

## 3. Results

### 3.1. Clinical History

The patient was referred to the Cardiology Service of Hospital Universitario 12 Octubre when he was 37 years old after presenting with dilated cardiomyopathy, probably secondary to the acute myocarditis at 25 years of age. The patient also presented with left renal atrophy with severe left renal artery stenosis of the celiac trunk and the superior mesenteric artery. After cardiological evaluation and due to the comorbidities he presented, he was referred to the Minority Diseases Service for clinical evaluation.

Physical examination at the age of 39 years revealed a height of 186 cm and a weight of 72 kg, corresponding to an asthenic phenotype. On clinical examination, the patient presented with emotional lability, certain shyness with little habitual contact, open mouth with frequent drooling, elongated face with a large forehead and large and low-set pinnae, joint hypermobility in the upper limbs and subtle kyphosis.

Despite the patient presenting with ID, the broad clinical signs were suggestive of a microdeletion/microduplication syndrome. Therefore, genetic testing for CNVs was requested.

### 3.2. Genetic Analysis

The CMA analysis revealed the presence of a 60 kb hemizygous deletion on Xq27.3 encompassing the entire *FMR1* gene with boundaries at 146,990,647 and 147,058,715 bp (arr[hg19] Xq27.3(146990647_147058715)x0). The deletion of the *FMR1* gene was further confirmed by an MLPA (Figure 2). The MLPA showed an absence of signal for all the probes that hybridize on the *FMR1* gene, confirming deletion of the entire gene.

The patient’s mother (I.1) was a phenotypically and cognitively normal woman. She underwent a CMA, detecting the same deletion in the heterozygous state with an intensity signal ratio of 0.49 below the normal range (Figure 3A). The expected ratio for a normal heterozygous deletion using this CMA is close to –1, suggesting an *FMR1* gene deletion in mosaicism. In order to validate this likely somatic mosaic, *FMR1* locus PCR amplification was performed using an AmplideX^®^
*FMR1* PCR assay in the mother. This assay revealed two normal alleles of 24 and 29 CGG repeats (Figure 3B). However, the intensity of the electrophoretic signal corresponding to the 24 CGG allele was abnormally underrepresented compared to that of the 29 CGG allele. In particular, the area of the 24 CGG repeat allele was 0.38× fold compared to the 29 CGG repeat allele (15,845 vs. 41,862; see Figure 3B). This atypical pattern confirmed the presence of an *FMR1* gene deletion approximately in 40% of the mother’s *FMR1* allele with 24 CGG repeats.

Although no genetic testing was performed on the parents of the mother, the fact that no other relatives are affected in the family suggests that the deletion developed as a postzygotic event in the mother (I.1). Nevertheless, the transmission of the deletion to her offspring (II.1) indicates that it was also present in the germinal cell line. Therefore, we extended genetic testing to the cognitively normal sibs of the proband (II.2 and II.3) (both—normal CMA). These results indicate that both sibs inherited the non-mosaic *FMR1* allele, albeit since a CGG repeat analysis has not been performed, we cannot determine which of the alleles (24 or 29) was transmitted.

## 4. Discussion

Mosaicisms in FXS usually refer to two different *FMR1* alterations, size and methylation mosaicisms [8]. The most common is “size mosaicism”, detected in 38% of the patients, which results in a variable clinical phenotype depending on the proportion of cells that have normal premutation alleles (55–200 CGGs) or FM alleles [9]. In addition, there are numerous cases of “methylation mosaicism”, in which an FM allele is fully or partially methylated. It has been observed that patients with this type of mosaicism are less severely cognitively affected as indicated by a less severe ID rating, higher intelligence quotient, higher adaptive behavior score, and lower social impairment score [10]. Although males with mosaic FXS have been generally reported to have better intellectual functioning, it has been described that FXS patients mosaic for PM and FM alleles have more severe irritability symptoms and maladaptive behaviors compared to males with only FM alleles associated with elevated *FMR1* mRNA levels that may be toxic in some cells [11]. However, very little is known about clinical manifestations associated with mosaicisms in FXS non-carriers of FM alleles.

To our knowledge, there are only three male cases reported in the literature caused by a mosaic of an *FMR1* gene deletion (Table 1). Hirst et al. (1995) reported an 8-year-old boy with a 678 bp deletion in 40% of peripheral lymphocytes that included the *FMR1* promoter region and the putative mRNA start site [12]. The patient had normal early development with nocturnal epileptiform seizures, but, at the time of referral, he presented with learning and behavioral difficulties, aggressive behavior, and the physical examination revealed obesity, macrotia, normal-sized testes, and macrocephaly (> 97th percentile) [12]. Coffee et al. (2008) reported an 11-year-old boy with a mosaic deletion of the entire *FMR1* gene in approximately 90% of his lymphocytes [13]. The patient showed mild ID, moderate to severe social anxiety, attention deficit/hyperactivity disorder-like symptoms, but facial findings distinctive of FXS were absent [13]. Finally, Hwang et al. (2016) reported a 33-year-old male presenting with a complex *FMR1* genotype consisting of unmethylated and methylated FM alleles as well as a *de novo* unmethylated microdeletion encompassing part of the *FMR1* promoter and the entire CGG repeat [14]. This patient presented with FXS with a milder than expected cognitive and behavioral impairments and white matter changes and progressive deterioration in gait with cerebellar signs consistent with probable fragile X-associated tremor/ataxia syndrome (FXTAS) [14]. On the basis of these observations, we can postulate that, as described for size and methylation mosaicisms, the mosaicism of an *FMR1* gene deletion is likely to cause a variable phenotype compared with the equivalent constitutive mutation due to the presence of a proportion of cells with the *FMR1* promoter active capable of producing FMRP.

On the other hand, there are only three reports of males with FXS caused by maternal mosaicisms of *FMR1* gene deletions (Table 1). All the three women were phenotypically unaffected. Prior et al. (1995) reported an FXS family with two sons with an *FMR1* deletion caused by a germline mosaicism in the healthy mother who did not carry the mutation in lymphocytes [15]. Luo et al. (2014) reported a healthy mother of a boy with FXS in whom the deletion was presented in four out of the 1000 cells analyzed by FISH [14]. Quantitative PCR analysis showed that she presented with varying proportions of the mosaicism among multiple tissues (4% in blood, 8% in skin fibroblasts, 11% in the urine sediment, 12% in the menstrual discharge, and 33% in the eyebrows) [16]. Jiraanont et al. (2016) reported a male patient with the typical features of FXS carrying a complete deletion of the *FMR1* gene whose mother carried the deletion in three out of the 400 metaphases from peripheral blood lymphocytes analyzed by FISH [17].

In our family, genetic analysis suggested that the deletion is present in approximately 40% of the mother’s cells in blood, representing a higher percentage compared to the previously identified families. In concordance with previous reports, the mother of our patient was phenotypically and cognitively normal. Therefore, the presence of the deletion in mosaicism together with the reduced penetrance associated with FXS in females might explain the absence of clinical features in women carrying *FMR1* gene deletion mosaicisms.

The increasing number of case reports of germinal mosaicisms has made geneticists more cautious with genetic counseling. Amounting evidence suggest that low-level (<10%) parental somatic mosaicisms for CNV deletions and point mutations are found in approximately 4% of the children affected by various genetic conditions [18,19]. The low levels of *FMR1* gene deletions reported in blood may be easily misdiagnosed as de novo transmission in routine genetic tests. Geneticists must be aware of this serious risk, taking into account that *de novo* mutations have been proposed as the major cause of many rare genetic disorders and, specifically, of neurodevelopmental disorders such as ID or ASD [20]. In clinical practice, whole blood samples typically undergo segregation analysis. However, mosaic variants might be underrepresented in the DNA derived from blood due to clonal expansion. Thus, the evidence herein compiled supports the recommendation of prenatal testing for apparently de novo mutations in sporadic FXS families. Detailed, in-depth parental profiling is necessary for accurate risk assessment in families with clinically defined *de novo* mutations.

## Figures and Tables

**Figure 1 genes-13-01609-f001:**
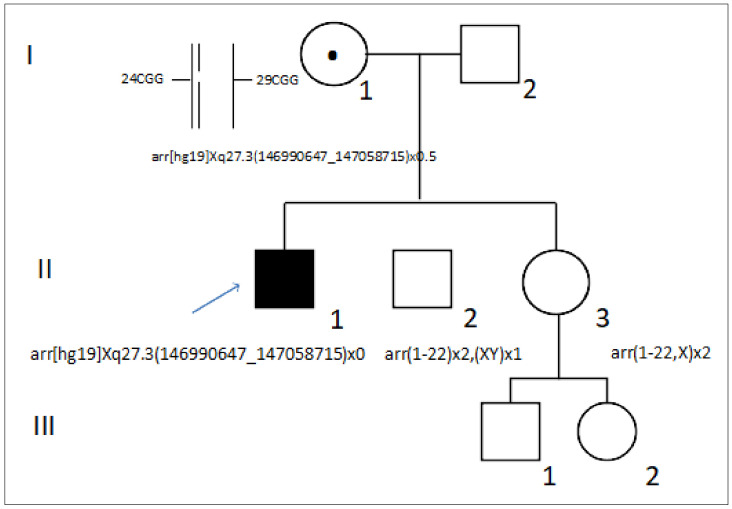
Family pedigree. The circles indicate females and the squares indicate males. The arrow indicates the index case. The black filled symbol indicates the patient presenting with ID. The dot indicates the female carrier of an *FMR1* gene deletion. *FMR1* genotypes are indicated for all the individuals tested.

**Figure 2 genes-13-01609-f002:**
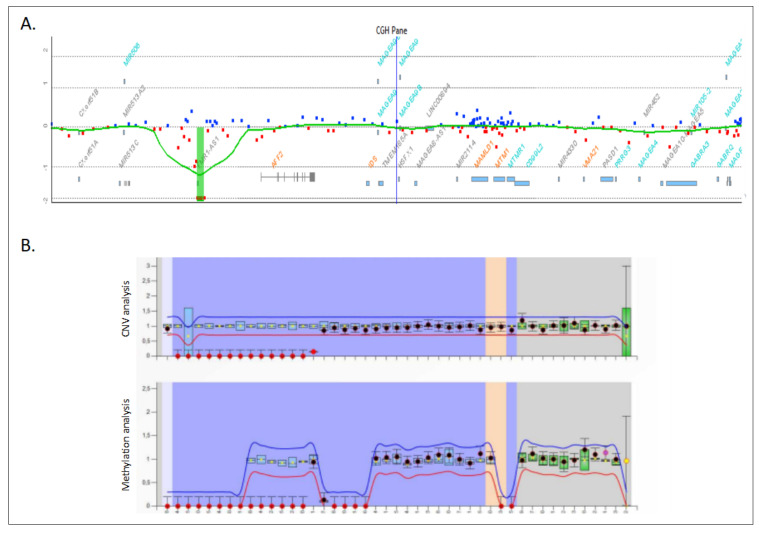
Genetic analysis of the index case. (**A**) *FMR1* gene deletion detected by a chromosomal microarray. (**B**) Validation of the *FMR1* gene deletion by means of an MLPA using SALSA ME029.

**Figure 3 genes-13-01609-f003:**
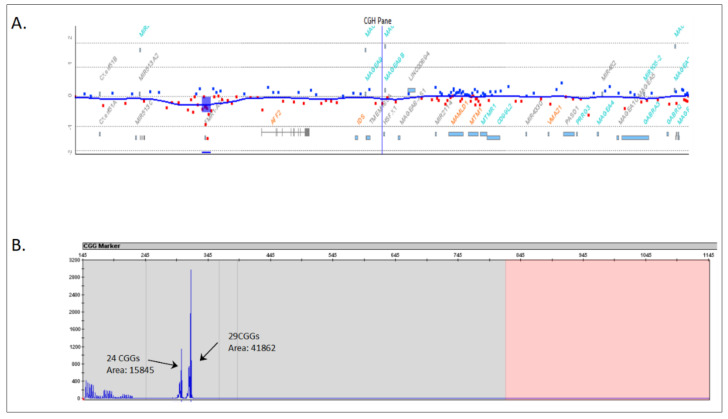
Genetic analysis of the mother (index case). (**A**) *FMR1* gene deletion detected by means of a chromosomal microarray. (**B**) Validation of the *FMR1* gene deletion mosaicism by means of an AmplideX^®^ FMR1 PCR assay. CGG repeat size and area of the peaks are indicated next to each other.

**Table 1 genes-13-01609-t001:** Individuals with *FMR1* gene deletions in mosaicism.

Sex	Clinical Features	Percentage in Blood	Genomic Coordinates (hg38)	Size	Genes Included	Ref
Male	Seizures, learning and behavioral difficulties	40%	X:147911457–147912135	678 bp	*FMR1* promotor	[12]
Male	Mild ID	90%	X:147158486–148171878	~1 Mb	*FMR1, FMR1-AS1*	[13]
Male	FXS with a milder cognitive impairement and FXTAS	NP	NP	~2.5 Kb	~80 bp of the *FMR1* promotor and the entire CGG repeat	[14]
Female	Unaffected	Undetected	NP	NP	*FMR1*	[15]
Female	Unaffected	0.4%	NP *	~45 Kb	*FMR1*	[16]
Female	Unaffected	0.75%	X:147653688–147955394	~300 Kb	*FMR1, FMR1-AS1*	[17]
Female	Unaffected	40%	X:146990647–147058715	~70 Kb	*FMR1, FMR1-AS1*	I.1

* Proximal: 5 kb upstream the *FMR1* gene; distal: 194 kb downstream the *FMR1* gene. NP: not provided.

## Data Availability

Genetic data from this family are available in the Genetic Service of Hospital 12 de Octubre upon reasonable request.

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
