# Peer review of "Fragile X Syndrome Caused by Maternal Somatic Mosaicism of FMR1 Gene: Case Report and Literature Review"

_genes, 2022, doi:10.3390/genes13091609_

Round 1

Reviewer 1 Report

 This is a relatively straightforward case report describing a single individual who inherited an X chromosome with a large deletion that spans the FMR1 gene thus resulting in the symptoms of Fragile X syndrome. The mother had two normal alleles but was mosaic for the deletion which occurred on the X chromosome containing the smaller of the two alleles. While a number of cases of children with FMR1 deletions have been described, relatively few originate from a mother who is mosaic for the deleted allele.

However, the authors seem to imply in various places that the clinical manifestations of patients who inherit a deletion from a mosaic parent may differ from those that inherit a deleted allele from a non-mosaic parent. Without a rationale for such a hypothesis, these statements should be removed.

The authors show a four generation pedigree in Fig. 1, yet from the description it seems that only the proband, his sibling and his mother were tested. This should be made clear in the diagram and/or its legend.

The authors note that 2.6 times more of the larger of the 2 maternal alleles is seen in blood. However, the authors then make the claim that this suggests that the deletion is present in 25% of maternal lymphocytes, when in fact it suggests that only ~25% of lymphocytes carry the normal allele.

The discussion would be improved if a diagram illustrating the endpoints of the deletion relative to other cases of deletions were provided. This would help the reader appreciate how and why some of these cases may show different clinical presentations. It would also be interesting to know if the endpoints of the deletions are associated with specific genomic features that could explain the deletion, for example endpoints located in interspersed repeats that might indicate some sort of intrachromosomal recombination event or the existence of recurrent deletion breakpoints.

There are two additional statements that need to be corrected.

Lines 40-41: The authors state: “Males are generally more affected, while females tend to present with a less severe phenotype due to compensatory activation of the unaffected X chromosome”

Since women have 2 X chromosomes that undergo random X inactivation in early embryogenesis, they are generally less affected than males because they express the deleterious allele in only a fraction of their cells. The unaffected X chromosome is not activated to compensate for the deleterious allele.

Lines 43-44: The authors correctly acknowledge the fact that the full name of the FMR1 gene has been changed to “Fragile X messenger ribonucleoprotein 1”. However, they then go on to use the old nomenclature for the protein product, FMRP, which has also been renamed (https://www.fraxa.org/fmr1-renamed-to-fragile-x-messenger-ribonucleoprotein-1/)

Author Response

We would like to thank the reviewer's comments which have improved the quality of the manuscript.

 This is a relatively straightforward case report describing a single individual who inherited an X chromosome with a large deletion that spans the FMR1 gene thus resulting in the symptoms of Fragile X syndrome. The mother had two normal alleles but was mosaic for the deletion which occurred on the X chromosome containing the smaller of the two alleles. While a number of cases of children with FMR1 deletions have been described, relatively few originate from a mother who is mosaic for the deleted allele.

However, the authors seem to imply in various places that the clinical manifestations of patients who inherit a deletion from a mosaic parent may differ from those that inherit a deleted allele from a non-mosaic parent. Without a rationale for such a hypothesis, these statements should be removed.

We apologize for this confusion. Individuals with germline deletions inherited or not from a mosaic parent are phenotypically identical to FXS syndrome patients with full mutation alleles. However, after reviewing the literature those males with the deletion of FMR1 gene in mosaicism are likely to present a less severe phenotype. We have modified the discussion in order to clarify this point.

The authors show a four generation pedigree in Fig. 1, yet from the description it seems that only the proband, his sibling and his mother were tested. This should be made clear in the diagram and/or its legend.

The pedigree has been simplify according to the reviewer suggestion. References in the manuscript have been modified accordingly.

The authors note that 2.6 times more of the larger of the 2 maternal alleles is seen in blood. However, the authors then make the claim that this suggests that the deletion is present in 25% of maternal lymphocytes, when in fact it suggests that only ~25% of lymphocytes carry the normal allele.

The genetic analysis performed to the mother of the index case by CMA and CGG TP-PCR revealed a normal FMR1 allele with 29 CGG repeats and a mosaic of a wt/deleted FMR1 allele with 24 CGG repeats.

The ratio of the deletion by CMA was 0.49 and the ratio of the smaller allele with the deletion was 0.4 respectively the 29 CGG wt allele by TP-PCR. Thus, suggesting that the deletion is present in approximately 40% of the FMR1 alleles with 24CGG repeats and thus in 20% of the mother lymphocytes.

This paragraph has been modified in order to clarify this point.

The discussion would be improved if a diagram illustrating the endpoints of the deletion relative to other cases of deletions were provided. This would help the reader appreciate how and why some of these cases may show different clinical presentations. It would also be interesting to know if the endpoints of the deletions are associated with specific genomic features that could explain the deletion, for example endpoints located in interspersed repeats that might indicate some sort of intrachromosomal recombination event or the existence of recurrent deletion breakpoints.

According to the reviewer suggestion a table with the molecular and clinical data of the individuals reported with FMR1 gene deletions in mosaicism has been included.

There are two additional statements that need to be corrected.

Lines 40-41: The authors state: “Males are generally more affected, while females tend to present with a less severe phenotype due to compensatory activation of the unaffected X chromosome”

Since women have 2 X chromosomes that undergo random X inactivation in early embryogenesis, they are generally less affected than males because they express the deleterious allele in only a fraction of their cells. The unaffected X chromosome is not activated to compensate for the deleterious allele.

We have modified this sentence according to the reviewer’s comment

Lines 43-44: The authors correctly acknowledge the fact that the full name of the FMR1 gene has been changed to “Fragile X messenger ribonucleoprotein 1”. However, they then go on to use the old nomenclature for the protein product, FMRP, which has also been renamed (https://www.fraxa.org/fmr1-renamed-to-fragile-x-messenger-ribonucleoprotein-1/)

We have modified the name of FMRP according to the new nomenclature.

Reviewer 2 Report

This is a short, concise manuscript that documents an increasingly important consideration in diagnosing a de novo event in a syndromic genetic disorder, that of cryptic maternal mosaicism.  This is a well documented consideration in the referenced 2020 paper of Lupski and coworkers (17).  In this instance the case deals with FMR1 and fragile X syndrome.  They document an FMR1 deletion in the propositus male, and demonstrate heterozigosity in mom, with the pathogenic allele carried at a greatly reduced abundance, based upon standard CGG repeat size markers.  Two neurotypical sibs are reported "normal" on a CMA but oddly the standard FMR1 markers are not reported.  It would be critical to know that they, presumably, carry the FMR1 allele associated with the non-mosaic X.  This should be explicitly reported in the paper.  A number of other family members are not genotyped, and are only inferred based on phenotype.  It would be helpful to the case to clearly have this presented, but that is not critical.  In view of the serious risk that a missed maternal mosaic poses to X-linked diseases, it is well worth emphasizing to readers that this is not an issue only confined to CMA syndromes, but that even monogenic disorders warrant this increased level of analysis required to detect low-level mosaicism.

Minor:  there are a few pronoun errors (L62) and other minor grammar errors that should be corrected, but overall the text is clear and well presented.  The figures are appropriate, with the caveat that the FMR1 CGG genotypes should be presented.

Author Response

We would like to thank the reviewer's comments which have improved the quality of the manuscript.

This is a short, concise manuscript that documents an increasingly important consideration in diagnosing a de novo event in a syndromic genetic disorder, that of cryptic maternal mosaicism. This is a well documented consideration in the referenced 2020 paper of Lupski and coworkers (17). In this instance the case deals with FMR1 and fragile X syndrome. They document an FMR1 deletion in the propositus male, and demonstrate heterozigosity in mom, with the pathogenic allele carried at a greatly reduced abundance, based upon standard CGG repeat size markers.

Two neurotypical sibs are reported "normal" on a CMA but oddly the standard FMR1 markers are not reported. It would be critical to know that they, presumably, carry the FMR1 allele associated with the non-mosaic X. This should be explicitly reported in the paper. 

This statement has been added in the results section.

A number of other family members are not genotyped, and are only inferred based on phenotype. It would be helpful to the case to clearly have this presented, but that is not critical.

The familial pedigree has been modified following the reviewer's suggestion.

In view of the serious risk that a missed maternal mosaic poses to X-linked diseases, it is well worth emphasizing to readers that this is not an issue only confined to CMA syndromes, but that even monogenic disorders warrant this increased level of analysis required to detect low-level mosaicism.

We have modified the discussion of the manuscript to address this point. A new reference has been included.

Minor: there are a few pronoun errors (L62) and other minor grammar errors that should be corrected, but overall the text is clear and well presented.

We apologize for these mistakes. The grammar error in L62 has been corrected and all the manuscript has been revised.

The figures are appropriate, with the caveat that the FMR1 CGG genotypes should be presented.

The FMR1 CGG genotype has only been determinate in the mother’s proband. However, all have CMA analysis. This information has been added to the figure 1.
